# Regional Analysis of Tracer Tests in the Karstic Basin of the Gacka River (Croatian Dinaric Karst)

Andrej Stroj ⬤, Maja Briški *⬤, Jasmina Lukač Reberski *⬤ and Tihomir Frangen

Department of Hydrogeology and Engineering Geology, Croatian Geological Survey, Sachsova 2, 10000 Zagreb, Croatia; astroj@hgi-cgs.hr (A.S.); tfrangen@hgi-cgs.hr (T.F.)
* Correspondence: mbriski@hgi-cgs.hr (M.B.); jlukac@hgi-cgs.hr (J.L.R.)

**Abstract:** Tracer testing is the only method in karst hydrogeology that can definitively determine whether a particular site belongs to a watershed of a particular karst spring. Therefore, it is an essential technique for delineating groundwater basins in karst areas. The availability of tracer test results is often limited due to the complicated and relatively expensive application of this approach, especially for large regional watersheds. The Croatian part of the Dinaric karst region extends for several hundred kilometers along the Adriatic coast and consists almost entirely of highly karstified carbonate rocks. The groundwater basins in these areas almost never match the surface morphology of the terrain. In practice, all available results of previous surveys are often used to define watersheds, regardless of the methodology and age of their implementation. This is also true for the earlier delineations of the Gacka River watershed, a regional karst basin in the Croatian Dinaric karst. However, tracer testing methods, especially the accuracy of tracer determination and monitoring, have improved significantly during this time. In order to assess the reliability of past tracing results in this significant karst basin, we reviewed reports of previous tracer tests. More recent tests, in particular the most recent multitracer injection test with continuous tracer detection on the major springs, produced high-quality data that allowed us to assess the reliability of the findings from prior research. A number of large karst springs with partially overlapping subcatchments feed the Gacka River. After discarding unreliable tracing data, we reevaluated the subcatchments of the main springs as well as the characteristics of the regional groundwater flow patterns throughout the basin, which is particularly important for water quality protection measures of the springs. The Gacka River basin is used as a case study to emphasize the importance of thoroughly assessing the reliability of previous tracing data before using them in regional analyses.

**Keywords:** tracer test; karst hydrogeology; catchment delineation; Dinaric karst; Gacka River

## 1. Introduction

The duality of groundwater flow in karst aquifers is the main feature that distinguishes them from other types of aquifers. It is a consequence of the gradual development of highly conductive conduit systems in the subsurface during the karstification process. Karst aquifers are therefore hydrogeologically conceptualized as a dual system with fast groundwater flow concentrated in conduits, which are surrounded by a low permeable fractured rock where groundwater only slowly percolates. These two media of hydraulically contrasting properties are rarely equilibrated and in constant interaction. Generally, the geometry of conduits controls regional groundwater flow direction, while slow draining of fractured rock mass provides water to maintain the baseflow on karst springs during drought periods. Although geological structures and relief can indicate flow direction, the exact locations of conduits are usually unknown. The inability to directly assess and define the geometry and position of conduits in the underground presents a major problem for detailed characterization, especially distributive flow modelling of regional karst systems.

Tracer testing is the only method that directly reveals underground flow routes and their characteristics. It is, therefore, the most important method to delineate karst spring

catchment, as well as to study flow and transport processes within [1]. A tracer test is commonly performed by injecting an artificial tracer directly into an active ponor (swallow hole), bypassing surficial soil and epikarst zone, to investigate local or regional conduit geometry and flow properties [2–9]. On the other hand, a tracer can also be dispersed on the terrain surface in order to examine infiltration processes together with epikarst and vadose flow characteristics [10–12]. Groundwater flow velocity (and sometimes also direction) is usually highly variable depending on hydrologic conditions, so in order to study an area in detail, it is desirable to perform multiple tests in various hydrological conditions. In addition, the point nature of tracing tests requires numerous tests to be performed across the karstic terrain in order to define catchment boundaries in detail, while groundwater flow bifurcations in conduit systems often result in significant overlapping of neighboring karst catchments. However, regional large-scale tracing tests are quite expensive and challenging to perform, so tracing results are often too scarce for precise delineation and reliable regional characterization of the catchments in complex karst terrains. This is especially true in the case of the Dinaric karst region which spreads along the eastern Adriatic coast across several countries from Italy, in the northwest, across Slovenia, Croatia, Bosnia and Hercegovina, and Montenegro to Albania. Dinaric karst is dominantly composed of highly karstified carbonate rocks with only minor inlays of nonkarstic terrains, so surface morphology and geologic structure most often do not provide much information on the catchment boundaries.

Since the earliest tracing tests, which were conducted in Croatian karst even before World War 2, tracing test procedures, particularly methods for tracer identification and monitoring, have significantly improved. Tracer testing in Croatian karst intensified within the framework of extensive regional hydrogeological investigations during the 1960s and 1970s [13]. However, up until the 1980s, the majority of fluorescein tracer detection in samples was done visually (with a UV lamp), and after that, fluorescence spectrophotometers became more often utilized. This significantly improved the detection limit and reliability of results determined as weakly positive by older methods. Namely, visual (UV lamp) detection of strongly positive samples is mostly reliable, but weakly positive samples can be false as the background from the presence of algae or other organic matter can be incorrectly interpreted as fluorescein [14]. This is especially the case when there is elevated turbidity in the sampled water, which is typical for many karst springs during high water conditions. Intense algal growth in the spring pool is also common during the growing season, which can greatly increase the fluorescence background of spring water. Therefore, a reliability assessment of the older tracing results is essential before further hydrogeological interpretation, considering the maximum estimated tracer concentration and the shape of the tracer breakthrough curve. However, during spring catchment delineation, often all available tracing results are considered without proper critical analysis and discarding of unreliable results. This can result in the wrong inclusion of some terrains within specific catchments, as well as in false information on expected groundwater velocities and residence time in karst systems.

The Gacka River flows from several abundant karst springs that drain the extensive karst basin with a total area of approximately 460 km$^2$. From the 1950s until today, a number of regional tracing tests were performed within the area. Delineation of the catchment and subcatchment boundaries, and their hydrogeological characterization, were largely based on the results of these tests [15]. However, after the recent performance of a simultaneous double tracer test on two injection points situated close to historical injection locations (active ponors), an inconsistency between recent and older results has been noticed. In addition, the new results provide a clearer picture of the underground flow pattern, in contrast to some hard-to-explain underground connections reported in historical tests. The main aim of this paper is to present recent tracing results in the Gacka River catchment, comparison and evaluation of older results, a new interpretation of groundwater flow characteristics after the rejection of groundwater connections that have been estimated as

unreliable, and, finally, to point out the importance of critical evaluation of tracing results reliability before their usage in hydrogeological characterization of an area.

## 2. Characteristics of the Area

The catchment area of the Gacka River is located in the western part of Croatia, which belongs to the wider area of the Dinaric karst. It is a hilly karst area where a large karst polje (Gacko Polje) is completely surrounded by the Velebit and Mala Kapela mountains. The Gacka River emerges from several major karst springs situated on the SE edge of the polje (Figure 1), flows across the mostly leveled polje surface, and sinks again in several ponors situated at the W and NW polje edges. Catchment of the springs ranges in elevation from 450 m a.s.l. (altitude of the main springs) to slightly above 1200 m a.s.l (elevation of the highest peaks within the catchment, Figure 1). Locally, relief plays a crucial role in the definition of climate conditions. The mean annual air temperature ranges between 4 °C and 9 °C [16], mostly controlled by the elevation of the terrain. Due to the vertical unevenness of the catchment, there is also a significant difference in the spatial distribution of precipitation. The annual mean precipitation values range between 1000 mm in the polje area and 2500 mm in the peak areas of the surrounding mountains [17]. The elevated terrains surrounding the Gacko polje are characterized by typical karst geomorphology, i.e., karst features are well expressed in the relief: numerous karrens and grykes, caves, dolines, and a few smaller karst poljes on higher elevation with intermittent springs and ponors—Vrhovine, Čanak, Trnavac, Homoljac, and Perušić poljes (Figure 1).

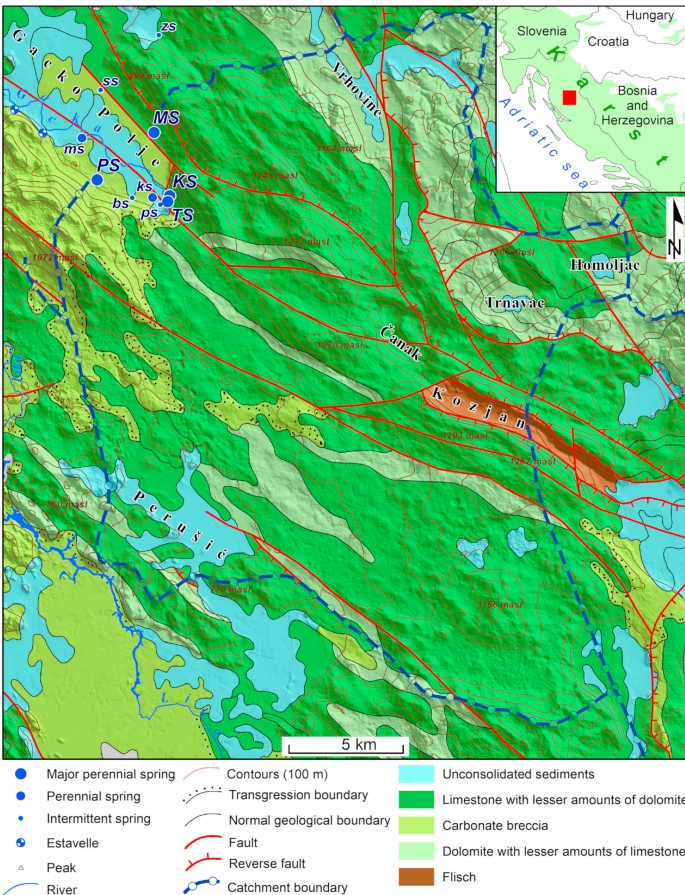

**Figure 1.** Hydrogeological map of the Gacka River watershed with the most important springs (Major springs: MS—Majerovo spring; KS—Klanac spring; TS—Tonković spring; PS—Pećina spring; other significant springs: ss—Sinac spring; ps—Pucirep spring; ks—Knjapovac spring; bs—Begovac spring; ms—Malinište spring; zs—Zalužnica spring).

The Gacka River's catchment area is almost exclusively composed of well-karstified carbonate rocks: Mesozoic limestones and dolomites and tertiary carbonate breccias. Both limestone and dolomite are prone to karstification, but the dissolution kinetics of dolomites is significantly slower [18]. As a result, there can be a significant difference between hydrogeologic properties and the function of limestone and dolomite terrains, but this is not uniform and depends on their position in the catchment. When limestone and dolomite rocks are in contact, the limestone area is more prone to karstification and usually becomes the preferred groundwater route. On the other hand, when dolomites form a continuous barrier between recharge and discharge area, they are also susceptible to karstification and allow the formation of fast conduit flows. In the Gacka River basin, this is confirmed by the results of the tracing tests, which are described in detail in the following chapters. On the terrain surface, dolomites are more susceptible to mechanical weathering and erosion, so a mellower terrain with less pronounced karst features is formed on them. However, caution is needed when evaluating the hydrogeological function of the terrain based solely on lithology. Tertiary breccias build the western parts of the karstic basin (the slopes of the Velebit Mt that separates the Gacko polje from the Adriatic Sea in the west) as well as the base of the Gacko polje. They consist of unsorted fragments of older carbonate rocks (mostly limestones) cemented by limestone material formed by crushing and breaking the same source rocks during the tectonically most active phase of the Dinaric Mountain formation (Eocene to Oligocene). Their compactness and susceptibility to karstification eventually lead to remarkable karst relief with strong dominance of corrosion over erosional landforms (e.g., karrens, grykes, and steep-sided solutional dolines).

Soil cover on the karst surface is mostly discontinuous, especially on limestones and carbonate breccias. On the other hand, the soil within karst depressions and as a filling of corrosion widened fractures (grykes) can be several meters thick. Intensely karstified epikarst allows a fast infiltration of precipitation, and surficial water flows are present only on karst polje floors. Leveled surfaces of karst poljes generally mark the height of the groundwater oscillation, with perennial or intermittent springs on upstream edges and ponors on the downstream ones (e.g., Čanak and Trnavac poljes). Some ravines are present on dolomitic hill slopes, which are formed by intermittent torrential water flows and are active only during very intensive precipitations. Fractures in the epikarst zone are mostly filled with clay, which together with soil cover increases the storage and mixing of infiltrated water. If the epikarst is fully saturated, or if the precipitation is too intense, excess water can bypass the epikarst storage and overflow directly into the conduit system. This is marked by the sudden changes in water chemistry at the Gacka springs after intense precipitation, mostly during extended wet periods when the epikarst is already saturated before the precipitation [19]. On the other hand, the spring water chemistry can remain stable, even after a relatively significant rain event after dry periods [20]. The storage and mixing capacity of epikarst in very compact limestone breccias, which are characterized by less soil cover on the terrain surface, appears to be lower than in limestones and dolomites. This is reflected by more a variable flow (very low baseflow) and the chemistry of karst springs recharged from terrains where breccias predominate (PS, Figure 1).

Unconsolidated alluvial, marsh, and lake sediments that cover the Gacka karst polje, are mostly low permeable. However, due to their variable thickness, which generally does not exceed a few tens of meters, they play a minimal hydrogeological function. In the downstream part of the polje, the Gacka River is occasionally "hanging", i.e., above the surrounding groundwater level. This is reflected by the appearance of ponors and estavelles at the contact of the polje sediments and carbonate rocks along the northwestern polje edges (outside the scope of Figure 1). The water that sinks into the ponors emerges again at the springs situated along the coast of the Adriatic Sea. Therefore, the Gacka River basin in its northwestern (downstream) part merges into a catchment of coastal karst springs and is separated from the catchment of the major springs of Gacka.

The Gacka River is fed by four major, and a number of secondary, springs (a few smaller perennials and a greater number of intermittent springs, Figure 1). The average annual

flow of the Gacka River is 14 m³/s, with the highest recorded maximum of approximately 70 and a minimum of 1.5 m³/s. The river flows across the polje surface from the spring area in the southeastern part of the polje to several ponors located along the northwestern polje edges. Four main springs, the Majerovo spring (MS), Kalanac spring (KS), Tonkovic spring (TS), and Pećina spring (PS), all discharge several to above ten cubic meters per second during high water periods. During dry periods, their discharge drops significantly but unevenly on particular springs. TS, which is used as a public water supply spring for the wider area, and MS are significantly more abundant springs in dry periods, when their discharge rarely falls below one cubic meter per second. On the other hand, KS's minimal annual discharge is usually a few hundred liters per second, and PS's discharge often falls below hundred liters per second. Therefore, MS and TS are the most important sources of the Gacka River during the dry season, while the majority of secondary springs dry up completely. There are only a few perennial secondary springs, all situated along the same fault line with the TS (ks and ms, Figure 1). As these springs show similar water chemistry as the TS [21], they are probably hydraulically well connected and share the same catchment.

All of the major springs exhibit limnocrene morphology [22]; this is in part due to man-made dams that served to build water mills in the past (Figure 2). Groundwater emerges from boulders and sediments at the bottom of spring lakes at all springs except the MS. The MS is located at the vertical outlet of a network of underwater channels and was explored and surveyed by cave divers over a length of more than 1000 m and to a depth of more than 100 m. The spring lakes are characterized by intense vegetation and algae growth that creates an elevated fluorescence background for tracer detection in the spring water.

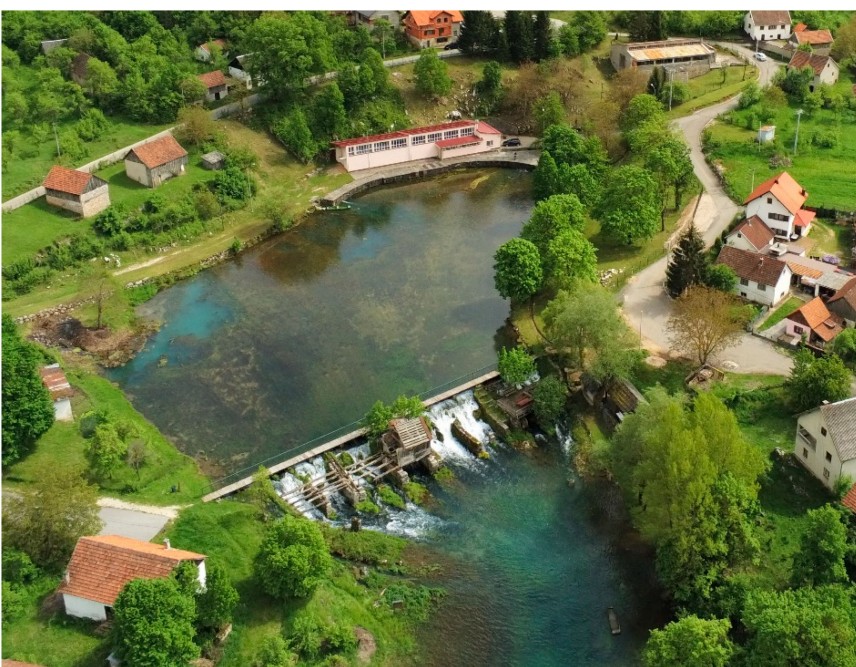

**Figure 2.** Tonković spring (TS) is characterized by typical limnocrene morphology, dispersed groundwater emergence from the sediments on the spring-lake bottom, and intensive freshwater vegetation.

## 3. Methods

Our research consisted of two steps: studying the literature of existing tracing tests and performing a new tracing test. In the first step, we collected and analyzed in detail the original technical documentation of all tracing tests conducted in the study area that showed underground water connections with the Gacka springs. Tracer samples of these former studies were collected manually, typically 3–4 times a day, while loggers for

continuous tracer monitoring were unavailable until the 2010s. During the 1970s, tracer detection was performed visually with the help of a UV lamp. Tracer concentrations were roughly estimated by comparison of the collected samples with the laboratory-prepared tracer solutions. Based on estimated concentrations, tracer breakthrough curves were constructed. From the 1980s on, tracer concentration was measured with the laboratory spectrophotometer, and from 2010 on, loggers for continuous monitoring were installed on the most important springs. We critically studied tracer concentration curves and the maximum recorded concentration, especially in the cases of visual tracer detection. For tracings where tracer detection was done visually, an estimated tracer concentration of 50 ppb or lower was assessed as potentially unreliable. Besides the maximum tracer concentration, we also reviewed the shapes of the tracer breakthrough curves in order to assess connection reliability and the time of the first arrival. In the cases of highly irregular curves with many peaks up to 50 ppb, intersected by lower concentrations, we rated the groundwater connection as false. Peaks higher than 50 ppb were generally evaluated as a reliable occurrence of tracers, but parts of the same breakthrough curve below 50 ppb were evaluated as unreliable. This, in addition to the rejection of some connections, also affected changes in the determined time of the first arrival, i.e., the maximum apparent velocity of the accepted underground water flows. In later tracing tests, we also examined data to detect potentially false connections due to turbidity and the high background fluorescence of spring water.

In the spring of 2019, we performed a simultaneous tracing test from the two ponors within the catchment. Traced ponors were located near the ponors traced in the 1970s in order to further check the reliability of the previous results, as well as to get more detailed quantitative data. Two tracers were injected into hydrologically active ponors in Vrhovine and Čanak poljes (Figure 1). At the time of injection similar flows of approximately ten liters per second were sinking into both ponors. We used relatively high amounts of tracers due to the high discharge of a large number of positive springs in previous tests: 30 kg of uranine was injected into the ponor in Vrhovine polje and 100 kg of Na-naphthionate into the ponor in Čanak polje.

The monitoring of the tracers was established at all active springs of the Gacka River. Four main sources (MS, KS, TS, and PS) were equipped with field fluorimeter loggers GGUN FL-30 by Abillia Sarl. The loggers enabled continuous tracer concentration monitoring as well as the monitoring of turbidity, which can affect tracer determination. In addition to the field fluorimeters, samples were also manually collected at all monitored springs. Tracer detection and concentration in the samples were determined in the laboratory on a Perkin-Elmer LS 55 fluorescence spectrophotometer. Besides the detection of the tracer arrival on minor springs, laboratory measurements also served for the control of the field fluorimeter measurements on the main springs. Blank samples were also taken before tracer injection in order to determine the fluorescence background of the spring water. Due to the relatively high background of the Gacka spring water, which is a consequence of the intense vegetation in the springs, the estimated minimum tracer concentration for reliable detection with the laboratory spectrophotometer was 0.01 ppb for uranine and 0.1 ppb for Na-naphthionate, and with the field fluorimeter approximately 0.2 ppb for uranine and 1 ppb for Na-naphthionate. Field fluorimeters measured tracer concentration every 15 min during the complete monitoring period. Water samples were collected twice daily during the first 20 days from the tracer injection, followed by once (or twice daily depending on hydrological conditions) afterward. Field fluorimeter concentration readings were additionally calibrated and adjusted to the values determined using a laboratory spectrophotometer. Monitoring finished 70 days after the injection, when tracer concentrations on all the springs were below the background.

At the four main springs, the discharge rate was also monitored continuously in addition to the monitoring of the tracer concentration. Discharge monitoring was established by continuous monitoring of the water level in the spring watercourses (HOBO water level logger), several flow measurements at various water levels (OTT MF Pro electromagnetic

current meter), and determination of the stage-discharge function on all measurement profiles. Occasional flow measurements were carried out at the minor springs also but without continuous level monitoring. Continuous discharge data enabled precise monitoring of the tracer recovery at the main springs.

## 4. Results

A detailed review of the original reports [23] revealed atypical and peculiar tracer breakthrough curves during both tracings carried out in the 1970s. The tracer concentration curves presented in the reports are characterized by numerous peaks of similar intensity irregularly intersected by the absence of a tracer (Figure 3). Additionally, visually estimated tracer concentrations, especially in peaks, seem to be largely overestimated in comparison with later tests with comparable injected tracer mass and hydrologic conditions (2019 test, Figure 3). After comparing results with the more recent tests (tracer concentrations were measured in the laboratory), most groundwater connections from the 1970s tracings were estimated as false. In more detail, all the results from the tracer testing of Kozjan polje were rejected, and only one connection from Vrhovine polje, the one toward the MS, was estimated as reliable. Groundwater connection from Vrhovine to MS was accepted due to the presence of a dominant peak of very high concentration as well as a good agreement with the results of the most recent tracing (Figure 3). However, we consider only the main peak of the curve to be a certain arrival of the tracer, while multiple preceding peaks are probably false. This results in a significant reduction of the apparent flow velocity of the first tracer arrival, which was reported as unusually high in the original report (9.2 cm/s). As the flow velocity of the first arrival is one of the main parameters for sanitary protection zone delineation in Croatian legislation, this has great importance for the determination of protection measures in the watershed. All the results of the later tracing tests were estimated as reliable, except one connection during the tracing from the doline near Perušić polje. Here, the tracer was detected on a ks on only a few water samples in relatively low concentration (<1 ppb) and correlated with turbidity peak. Prior to and after this, the water samples were negative, so tracer detection was probably a consequence of the turbidity-related background in the spring water. However, this connection does not change the regional interpretation of the groundwater flow directions significantly, unlike the rejection of the connections from the 1970s tracing tests.

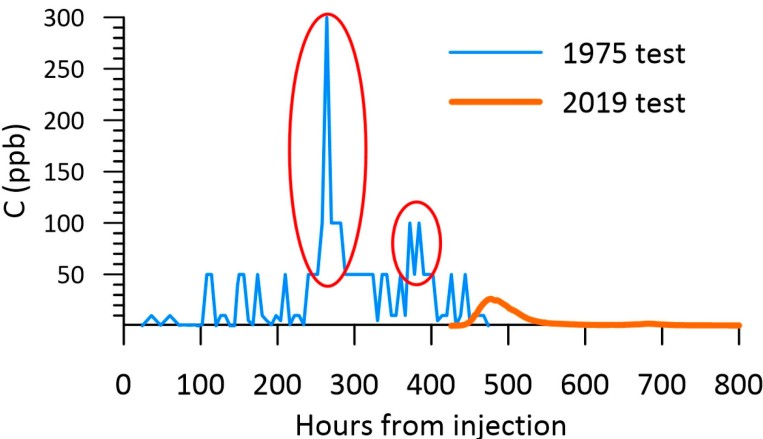

**Figure 3.** Comparison of the tracer breakthrough curves on MSs during tracing in Vrhovine polje in 1975 and 2019 (1975 concentrations were estimated visually, and we rated only parts of the curve marked with red circles as reliable tracer detections).

Table 1 shows the results with selected parameters of all tracer tests performed in the Gacka River watershed until today. There have been 12 tracing tests with 28 determined groundwater connections. After analyzing and rejecting some unreliable tracer detections, we estimated 21 groundwater connections as reliable. In most tests, the tracer was injected

in active ponors, while once the tracer was injected in an observational borehole near Perušić polje, once in the crack widened through corrosion in the quarry located near the hinterland of the springs, and once in the solutional opening in the rocky bottom of the doline near Perušić polje. After the injections at locations that were not active ponors, the tracer was additionally flushed with several tens of cubic meters of water. After rejecting unreliable results, all apparent flow velocities (both to first tracer arrival and peak concentration) were within the range of 0.2–4.4 cm/s, with a mean value of 1.7 and a median value of 1.3 cm/s. Apparent groundwater flow velocities are well correlated to concurrent Gacka River flows, i.e., with hydrological conditions by faster velocities in wetter periods (Figure 4). It should be noted that all tracing tests were done in moderate to high-flow conditions, so apparent groundwater flow velocities during low-flow conditions are probably significantly lower, i.e., below 0.1 cm/s. Generally, flow velocities can vary by even two orders of magnitude depending on the hydrological conditions in the karst conduits. The arrival of the tracer is often pushed during, or soon after, hydrograph peaks, so besides average flow conditions, flow dynamics after the injection also considerably affect tracer arrival time (continuous flow recession vs. flood event after the injection). On the other hand, there are no significant differences in apparent groundwater flow velocities between injections into active ponors and a few injections into other types of objects (Table 1), which is probably due to the selection of highly permeable locations and the washout of the tracer by a significant amount of water after injection.

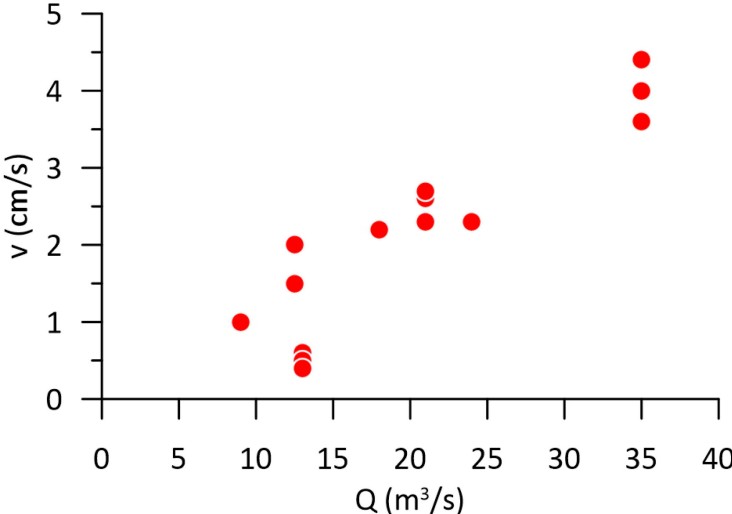

**Figure 4.** Dependency of apparent flow velocity on hydrological conditions, i.e., average Gacka River flow during the period between the injection and tracer maximum concentration (only tracing tests of active ponors, see Table 1).

**Table 1.** Tracing tests in the Gacka River catchment and the main parameters of detected arrivals: $C_{max}$—maximum tracer concentration in spring water; $v_{max}$—maximum apparent velocity; and $v_{Cmax}$—apparent velocity to maximum concentration [20,23–30], according to data from original reports.

| Injection Date | Injection Location | Tracer Arrival [1] | Distance (km) | $\Delta h$ (m) | Tracer/Mass (kg) | Gacka R. Q ($m^3/s$) [2] | $C_{max}$ (ppb) | Time to First Arrival (Hours/Days) | $v_{max}$ (cm/s) | Time to $C_{max}$ (Hours/Days) | $v_{Cmax}$ (cm/s) | Recovery (%) |
|---|---|---|---|---|---|---|---|---|---|---|---|---|
| 14.4.1957 | Perušić ponor 1 | PS | 10.9 | 108 | Uranine/30 | 24 | 25 | 110/4.5 | 2.8 | 134/5.6 | 2.3 | 30 |
| 13.1.1975 | Vrhovine ponor | zs | 9.7 | 227 | Uranine/50 | 9 | * 50 | 306/1.75 | 0.9 | 264/11.0 | 1 | ? |
|  |  | ss | 11.9 | 275 |  |  | * 10 | 66/2.75 | 5 |  |  |  |
|  |  | MS | 9.9 | 273 |  |  | * 300 | 30/1.25 | 9.2 |  |  |  |
|  |  | KS | 10.1 | 277 |  |  | * 50 | 42/1.75 | 6.8 |  |  |  |
|  |  | TS | 10.3 | 278 |  |  | * 50 | 42/1.75 | 6.8 |  |  |  |
| 21.3.1975 | Kozjan ponor | ms | 20 | 351 | Uranine/60 | 17 | * 10 | 263/11.0 | 2.1 |  |  |  |
|  |  | PS | 18.6 | 348 |  |  | * 5 | 281/11.7 | 1.8 |  |  |  |
|  |  | KS | 15.6 | 346 |  |  | * 5 | 197/8.2 | 2.1 |  |  |  |
| 9.12.1986 | Perušić borehole | PS | 9 | 164 | Uranin/30 | 12 | 60 | 255/10.5 | 1 | 288/12 | 0.9 | 74 |
| 8.4.1997 | Lešće quarry | ps | 1.25 | 215 | Uranine/6 | 12 | 70 | 27/1.1 | 1.3 | 33/1.4 | 1.1 | 48 |
|  |  | ks | 1.65 | 211 |  |  | 70 | 44/1.8 | 1 | 50/2.1 | 0.9 | 15 |
|  |  | bs | 1.98 | 213 |  |  | 0.8 | 116/4.8 | 0.5 | 186/7.8 | 0.3 | 6 |
|  |  | PS | 1.35 | 214 |  |  | 0.2 | 164/6.8 | 0.2 | 186/7.8 | 0.2 | 10 |
| 12.12.2002 | Perušić ponor 2 | PS | 12.5 | 106 | Uranine/34 | 18 | >100 | 136/5.6 | 2.6 | 160/6.6 | 2.2 | ? |
| 30.3.2010 | Trnavac ponor | TS | 16 | 263 | Uranine/5 | 21 | 0.5 | 154.5/6.4 | 2.9 | 170/7.1 | 2.6 | 3 |
|  |  | KS | 16 | 262 |  |  | 3.3 | 140/5.8 | 3.2 | 164/6.8 | 2.7 | 41 |
|  |  | MS | 17.7 | 258 |  |  | 0.8 | 164/6.8 | 3 | 212/8.8 | 2.3 | 18 |
| 23.4.2010 | Trnavac ponor | KS | 16 | 262 | Uranine/15 | 12.5 | 6.9 | 213/8.9 | 2.1 | 296/12.3 | 1.5 | 52 |
|  |  | MS | 17.7 | 258 |  |  | 4.7 | 264/11 | 1.9 | 312/13 | 2 | 20 |
| 12.3.2013 | Homoljac ponor | TS | 17.9 | 298 | Naphthionate /75 | 35 | 1.5 | 114/4.7 | 4.4 | 114/4.7 | 4.4 | 1 |
|  |  | KS | 17.9 | 297 |  |  | 17.1 | 114/4.7 | 4.4 | 126/5.2 | 4 | 14 |
|  |  | MS | 19.3 | 293 |  |  | 5.3 | 138/5.7 | 3.9 | 150/6.2 | 3.6 | 8 |

**Table 1.** *Cont.*

| Injection Date | Injection Location | Tracer Arrival [1] | Distance (km) | $\Delta h$ (m) | Tracer/Mass (kg) | Gacka R. Q $(m^3/s)$ [2] | $C_{max}$ (ppb) | Time to First Arrival (Hours/Days) | $v_{max}$ (cm/s) | Time to $C_{max}$ (Hours/Days) | $v_{Cmax}$ (cm/s) | Recovery (%) |
|---|---|---|---|---|---|---|---|---|---|---|---|---|
| 15.9.2014 | Perušić doline | PS | 12 | 132 | Uranine/25 | 33 | 0.4 | 48/2 | 6.9 | 343/14.5 | 1 | ? |
| | | ks | 10.6 | 129 | | | 0.5 | 162/6.7 | 1.8 | 170/7.1 | 1.7 | ? |
| 20.3.2019 | Vrhovine ponor | MS | 9.8 | 273 | Uranine/30 | 13 | 26.2 | 432/18 | 0.7 | 480/20 | 0.6 | 72 |
| 20.3.2019 | Čanak ponor | MS | 13.4 | 163 | Naphthionate /100 | 13 | 9.6 | 816/34 | 0.5 | 912/38 | 0.4 | 19 |
| | | KS | 11.2 | 167 | | | 10.4 | 792/33 | 0.4 | 888/37 | 0.4 | 32 |

[1] Major Gacka springs: MS—Majerovo s.; KS—Klanac s.; TS—Tonković s.; PS—Pećina s.; Secondary Gacka springs: zs—Zalužnica s.; ss—Sinac s.; ps—Pucirep s.; bs—Begovac s.; ks—Knjapovac s.; ms—Malinište s. [2] Average Gacka River flow during period between tracer injection and $C_{max}$ on spring, according to DHMZ data http://hidro.dhz.hr/ (accessed on 1 May 2023); * Visually estimated tracer concentration. Rejected tracer arrivals and parameters are marked in red.

In most of the tracer tests, uranine was used as a tracer, while Na-naphthionate was also used in some of the more recent tests. Recovered tracer mass in spring water has a range of 30–79% for uranine, and 23–51% for naphthionate. It seems that naphthionate is slightly less conservative tracer compared to uranine, but more results would be required for a reliable answer to this. In addition, it should be noted that a reliable estimation of recovered tracer mass strongly depends on accurate flow monitoring, which is generally less accurate in high-flow conditions compared to low-flow conditions. Maximum tracer concentration usually follows relatively quickly on the first arrival, while the falling limb of the tracer breakthrough curve is typically gentler with pronounced tailing, and secondary minor peaks are often present. This results in a typically small difference between the first arrival (or maximum) velocity and the maximum concentration (or dominant) velocity. However, the maximum concentration velocity can be determined more precisely, while the first detection of tracer arrival can be slightly arbitrary (dependent on instrument detection limit and spring water background fluorescence).

The most recent tracer test from two ponors (Vrhovine and Čanak poljes, Figure 1) provides the most detailed quantitative data due to continuous monitoring of tracer concentrations on all major springs and additional flow monitoring on all springs separately (Figure 5). Tracers were injected in active ponors (inflow rates of several l/s) at a flow rate of the Gacka River that was lower than all prior tracings (8 m$^3$/s), which is slightly below its average annual flow rate (14 m$^3$/s). Flow recession continued after the injections for 15 days, followed by a more humid period with several moderate hydrograph peaks. The first tracer arrival from the Vrhovine polje quickly followed the first flow peak, while the tracer from Čanak arrived 15 days later in the recession period, 8 days after the second (higher) flow peak. The tracer from Vrhovine arrived only to one spring (MS), and from Čanak to two springs (TS and MS). These results match well with the results of recent tracings from nearby locations (Trnavac and Homoljac, Table 1) and confirm the rejection of most of the tracing connections from the 1970s tracings. The tracer curve of uranine injected at Vrhovine shows significantly lower dispersion and higher tracer recovery in comparison to naphthionate from Čanak polje. All the tracer breakthrough curves show secondary minor peaks, and are most pronounced in the case of the tracer from Čanak on KS.

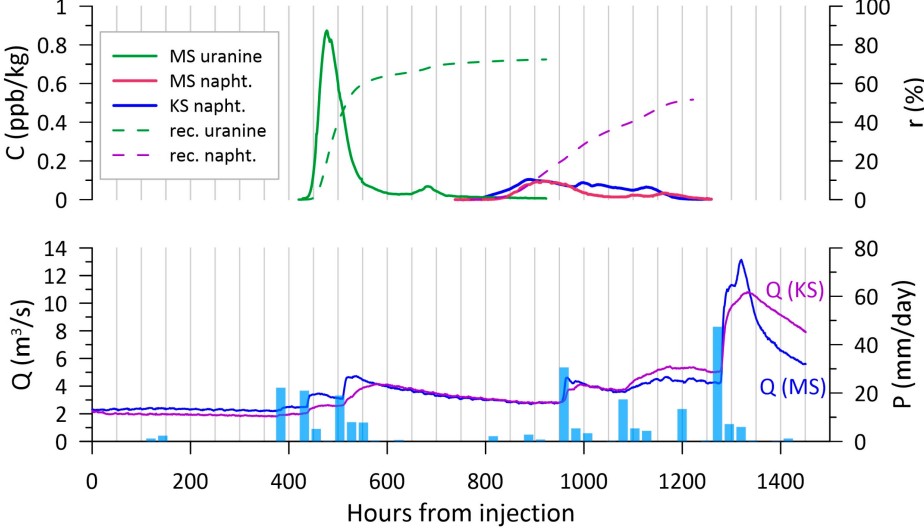

**Figure 5.** Tracer breakthrough and tracer recovery curves during a simultaneous tracing test from Vrhovine polje and Čanak polje, concurrent hydrographs of the MS and KS, and daily precipitations measured on the TS location. Tracer concentrations are normalized by injected tracer mass. Both KS and MS were equipped with field fluorimeters for continuous tracer monitoring.

## 5. Discussion

After the rejection of implausible groundwater connections, the main characteristics of underground flows in the karst basin of the Gacka River have become more clearly expressed. Underground flows from the more distant areas in the basin concentrate and flow toward the main springs, while secondary springs mostly drain areas in their closer hinterland (except ks and ms which are most likely fed from TS along the regional fault line which connects them, Figure 1). A tracing test from Lešće quarry, situated 1–2 km from the springs, is the only one that proves groundwater flow toward some secondary springs. All other tests from more distant locations have shown the flow directions toward the four major springs only (Figure 6). This is in accordance with the hydrological estimation that four major springs provide 70% of Gacka River flow on average and almost 100% during dry periods [20].

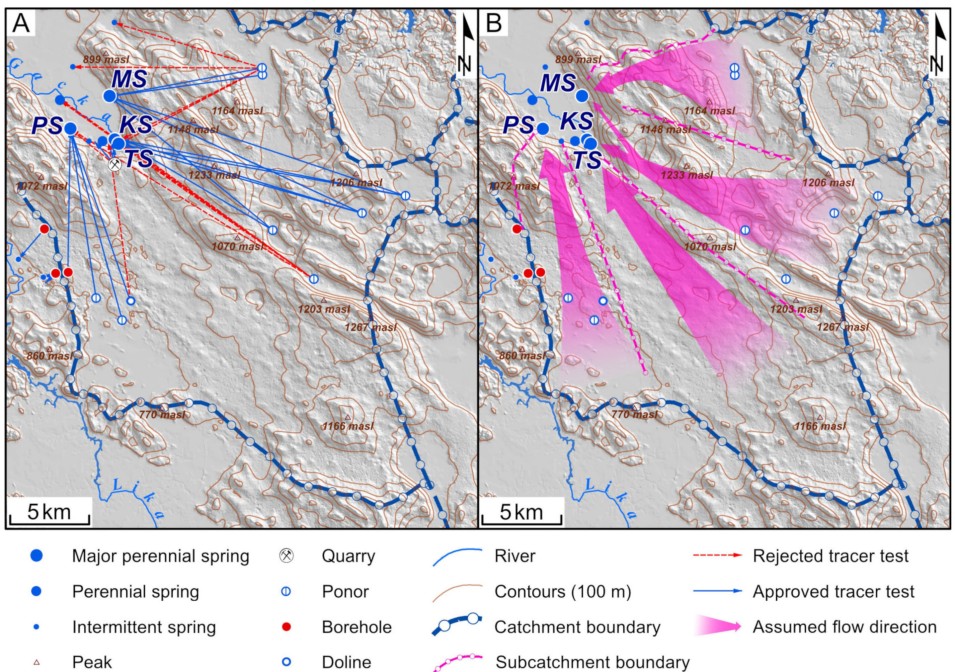

**Figure 6.** (**A**) Map showing accepted (blue lines) and rejected (red lines) groundwater connections; (**B**) interpretation of regional groundwater flow directions and subcatchment areas of the major springs after the rejection of unreliable results.

A strong correlation between apparent groundwater flow velocities and flow conditions (Figure 7) points to a significant volume of voids within conduit systems in the karst underground. The exchange of water volumes stored in conduits controls tracer travel time, which is in inverse correlation with flow fluctuations. The existence of spacious submerged chambers and voids is directly proved by cave diving exploration of channels that end in MS spring-lake. A mostly linear decrease of travel time with an increasing spring flow rate indicates a high connectivity of conduit systems without significant obstructions (e.g., crushed zones) that would slow the groundwater flow in all flow conditions. However, due to numerous but unknown underground confluences, i.e., groundwater flow concentration with multiple increases of flow rate along the path from the ponor to the spring, it is not possible to make a quantitative estimation of the volume of conduit systems based on the tracer travel time and the spring flow rate.

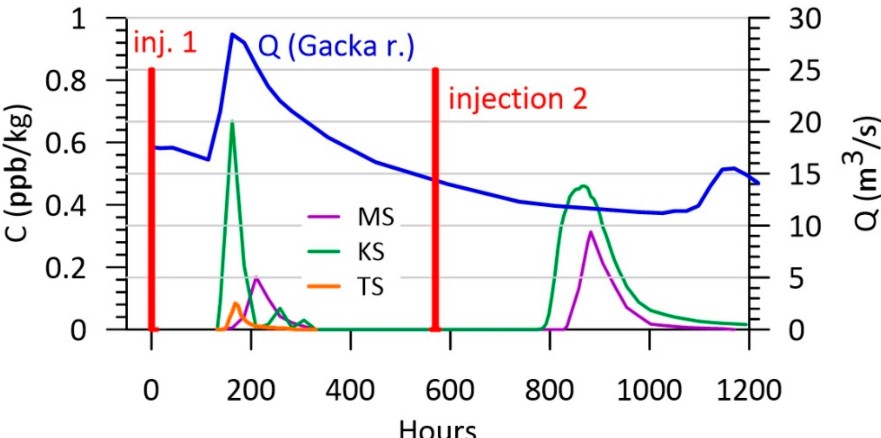

**Figure 7.** Tracer breakthrough curves during repeated tracer tests from Trnovac polje (tracer concentrations are normalized by injected tracer mass; the Gacka River hydrograph is shown to compare tracer curves with concurrent flow conditions).

Generally, the rejection of unreliable results also enabled a more consistent and clearer picture of the regional groundwater flow pattern. Overlapping of the major spring subcatchments is much less pronounced, and crossings of underground flow connections are much less widespread (Figure 6). Based on characteristics of accepted groundwater connections, as well as on the water budget of individual springs [20], it was possible to divide the catchment into four mostly separated subcatchments: the MS drains northern parts of the Gacka catchment independently (area of Vrhovine polje, Figure 1), while the KS and MS jointly drain Homoljac, Trnavac, and Čanak poljes and surrounding areas to the south. The tracings in Vrhovine, Trnovac, and Homoljac also showed that dolomites in this area do not create barriers or significantly slow down the groundwater flow (Figure 1). The southeastern portion of the entire Gacka catchment is covered by the largest subcatchment, which is drained by the TS (and a few secondary springs hydraulically connected with the TS). This subcatchment is the most poorly defined, considering that no tracer test was done from its area, which is mostly due to the absence of karst poljes and intermittently active ponors. However, tracing testing from this area is a priority for future investigations of the Gacka basin. The SE part of the Gacka basin, i.e., Perušić polje and its wider surrounding, is exclusively drained by the PS. A smaller amount of groundwater from the PS subcatchment probably drains underground directly toward the Adriatic Sea coast, without emergence at the PS, which would explain its very low flow in dry periods.

There is a special hydraulic relation between the TSs and KSs, which are situated only 200 m apart. Although according to tracer tests they have separate catchments, during high-flow conditions, some water from the KS can penetrate to the TS, which is reflected by the appearance of low-intensity tracer peaks at the TS, simultaneous with much higher tracer peaks at the KS (tracings of Trnovac and Homoljac ponors, Table 1). At times of more moderate flows, the tracer appears at the KS only, as can be seen in the case of repeated tracing of Trnovac ponor (Figure 7) and the tracing of Čanak polje (Table 1, Figure 5). On the contrary, during low-flow periods when the TS has a much higher flow rate than the KS, according to the chemical characteristics of the spring water [20], it seems that the KS receives water from the TS.

Finally, it should be emphasized that the boundaries of subcatchments are approximately positioned and that it is not possible to fully determine to what extent their position is variable depending on hydrological conditions, nor to what extent subcatchments permanently or intermittently overlap. The same applies to the outer limits of the entire Gacka River catchment. Nevertheless, the division of the basin into subcatchments greatly influences the establishment of protection measures, i.e., it is possible to determine in more detail which areas are critical for the protection of individual springs.

In the simultaneous tracing test of Vrhovine and Čanak poljes in 2019, the naphthionate tracer from Čanak showed lower maximum concentration with higher dispersion, lower apparent velocity, and lower total tracer recovery (Figure 5). This can be related to several factors: slightly longer underground flow path (35% longer distance from Čanak to the MS in comparison to Vrhovine to the MS); greater dilution of the tracer due to two major outflow points (MS and TS); and possibly a less conservative behavior of naphthionate compared to uranine. All the tracer breakthrough curves show secondary minor peaks, which seems to be related to spring hydrograph peaks (with some delay). Secondary peaks may be caused by the activation of low-mobile parts of the conduit system during rising flow, pushing the tracer stored therein, so they do not necessarily prove alternative groundwater flow paths. Generally, higher dispersion and tailing of tracer breakthrough curves in lower flow conditions (Figure 7) indicate a dominant control of conduit morphology (mobile and less mobile regions within them [3,31]) and a negligible role of tracer storage in surrounding low permeability rock mass [32]. This is also consistent with typical borehole characteristics from the area of the Croatian Dinaric karst, which normally have negligible yields if they do not intersect significant karst channels.

## 6. Conclusions

A critical review of older tracer test reports, based mostly on comparison with the results of more recent tests with more precise tracer detection, has shown that a substantial number of older results are not reliable.

The rejection of unreliable groundwater connections provided a clearer picture of the regional groundwater flow pattern in the Gacka River karst basin and allowed the delineation of four subcatchments. After rejection, tracer breakthrough curve characteristics and apparent groundwater flow velocities become also more consistent across the results.

Repeated tracings provide data on the groundwater flow variability under different hydrologic conditions, which provides better insight into the characteristics of the conduit systems. Hydrologic conditions are found to be the main factor affecting groundwater flow velocity, while velocities are relatively similar at different tracing locations under similar conditions. This proves the existence of well-developed conduit systems that connect all injection locations with the springs, while the limestone or dolomite lithology does not significantly influence flow velocities. The inverse relationship between tracer travel time and spring flow rate, as well as the higher dispersion of the tracer at lower flow rates, indicate that the volume and morphology of the conduits have the greatest influence on groundwater flow velocities and dispersion, while water exchange between the conduits and the surrounding rock has a minor influence.

For the estimation of the minimum possible travel time from ponor to the spring, which is the main parameter for the delineation of the karst groundwater protection zones in Croatia, it is important to inject tracers at high-flow conditions. At low flows, groundwater flow velocities are usually at least an order of magnitude slower. However, only additional tests in moderate and/or low water conditions can provide important information on groundwater flow variations under changing hydrologic conditions.

To accurately delineate the boundaries of catchments and subcatchments, tracer tests are required at numerous locations throughout the investigated area. In the case of the Gacka River basin, this is still not fulfilled, so catchment and subcatchment boundaries are approximately positioned in places, and significant deviations are still possible.

Finally, we believe that the described findings of the tracing test analysis in the Gacka River basin are valid and applicable to many other regional basins in classic karst areas worldwide.

**Author Contributions:** Writing of original draft, conceptualization, investigation, A.S.; investigation, analysis, visualization, writing—review and editing, M.B.; investigation, writing—review and editing, J.L.R.; investigation, analysis, visualization, writing—review and editing, T.F. All authors have read and agreed to the published version of the manuscript.

**Funding:** This research was supported by the Croatian Geological Survey, Department of Hydrogeology and Engineering Geology, through the program of the Hydrogeological Map of Croatia (project 181-1811096-3165); most of the tracer tests were financed by HRVATSKE VODE, a legal entity for water management.

**Institutional Review Board Statement:** Not applicable.

**Informed Consent Statement:** Not applicable.

**Data Availability Statement:** All data are available on request.

**Acknowledgments:** The authors wish to express gratitude to our colleagues who were involved in all prior tracer tests included in this paper.

**Conflicts of Interest:** The authors declare no conflict of interest.

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
