# Peer review of "Regional Analysis of Tracer Tests in the Karstic Basin of the Gacka River (Croatian Dinaric Karst)"

_hydrology, doi:10.3390/hydrology10050106_

Round 1

Author Response

Thank you for your positive opinion about the manuscript, and for all the useful comments that enabled further improvement of it. Please see the attachment.

Reviewer 2 Report

The difficulty in determining the boundaries of karst basins is the biggest challenge in the study of hydrogeological processes in these areas. This study is based on current relatively accurate tracer experiments, and compared with previous experimental studies, it is found that there are many problems in the past research results, which is of great significance for the study of hydrological processes in the study area and other karst areas in the world.

Author Response

Thank you for your positive review. We are glad that you find our work significant for the study of hydrological processes in karst areas.

Reviewer 3 Report

The manuscript with the title " Regional analysis of tracer tests in the karstic basin of the Gacka River (Croatian Dinaric karst)" by Stroj et al. has been reviewed. This manuscript in its present form should be accepted. It is suggested that the authors introduce some information such as topographic contours and water level contours into the paper and conduct a comparative analysis. It seems that studying a clearer picture of regional groundwater flow pattern in the Gacka River karst basin only from the perspective of tracer tests is not quite sufficient.

Author Response

Thank you for your positive review and useful comments.

We have added topographic contours on Figures 1 & 6. However, due to the high heterogeneity of the water table in studied karst basin, as well as the absence of boreholes in most of the basin area, we believe that it is not possible to reconstruct the water level contours reliably. The main information on the height of the karst water level is the height of intermittent and permanent springs, which appear only in the area of smaller karst poljes within the basin (Vrhovine, Čanak, Perušić). Height of this poljes is visible based on topographic contours on Figure 6, and we have also added the height difference between injection points and springs in Table 1.

We agree that tracing tests alone are insufficient for complete picture of groundwater flow characteristics in karst basin, especially regarding diffuse infiltration and storage processes. We have also performed monitoring and analysis of physiochemical and isotopic parameters on major Gacka springs. However, we focused on the tracing test aspect of hydrogeological research considering that the inclusion of other methods would unduly expand the scope of the paper, and is planned to be published separately.

Reviewer 4 Report

see attachment

Author Response

(The authors gave the same response as above.)
